# Robust and Reconfigurable On-Board Processing for a Hyperspectral Imaging Small Satellite

**Dennis D. Langer** [1,2,*] ![ID], **Milica Orlandić** [2,3], **Sivert Bakken** [2,4,5] ![ID], **Roger Birkeland** [2,3] ![ID], **Joseph L. Garrett** [2,5] ![ID], **Tor A. Johansen** [2,5] and **Asgeir J. Sørensen** [1,2]

1   Department of Marine Technology, NTNU, 7491 Trondheim, Norway
2   Center for Autonomous Marine Operations and Systems, Marine Technology Center, 7491 Trondheim, Norway
3   Department of Electronic Systems, NTNU, 7491 Trondheim, Norway
4   SINTEF Ocean, 7052 Trondheim, Norway
5   Department of Engineering Cybernetics, NTNU, 7491 Trondheim, Norway
*   Correspondence: dennis.d.langer@ntnu.no

**Abstract:** Hyperspectral imaging is a powerful remote sensing technology, but its use in space is limited by the large volume of data it produces, which leads to a downlink bottleneck. Therefore, most payloads to date have been oriented towards demonstrating the scientific usefulness of hyperspectral data sporadically over diverse areas rather than detailed monitoring of spatio-spectral dynamics. The key to overcoming the data bandwidth limitation is to process the data on-board the satellite prior to downlink. In this article, the design, implementation, and in-flight demonstration of the on-board processing pipeline of the HYPSO-1 cube-satellite are presented. The pipeline provides not only flexible image processing but also reliability and resilience, characterized by robust booting and updating procedures. The processing time and compression rate of the simplest pipeline, which includes capturing, binning, and compressing the image, are analyzed in detail. Based on these analyses, the implications of the pipeline performance on HYPSO-1's mission are discussed.

**Keywords:** cubesat; hyperspectral imaging; image processing system; hyperspectral compression; on-board processing

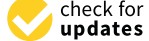

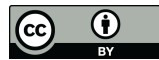

## 1. Introduction

Remote sensing satellites are an indispensable tool for Earth mapping and monitoring, and for improving the predictability of Earth system processes. Future remote sensing satellites that provide hyperspectral imagery promise to be a valuable source of information and enable new applications [1,2]. There have been about two dozen spaceborne hyperspectral imaging sensors [3], starting with the Earth Observing-1 (EO-1) and Project for On-Board Autonomy 1 (PROBA-1) missions in the early 2000s [4,5]. Further hyperspectral earth observation satellites continued to demonstrate the usefulness of hyperspectral data and many more are planned [6–13]. However, the infrequent nature of their observations limits the use of hyperspectral sensors in applications with strong temporal requirements, such as disaster monitoring or the study of continuous ecosystem dynamics.

Satellites with multispectral remote sensing instruments, such as those in the Landsat [14] and Sentinel [15] missions that record and downlink data continuously on the day side of the earth and rapidly provide data to end-users, have often been used for such applications. These satellites are able to provide rapid and temporally well-resolved data because, for a given area, they produce fewer data than hyperspectral imagers at a given spatial resolution. The challenges that hyperspectral missions face are exemplified by the upcoming Copernicus Hyperspectral Imaging Mission for the Environment (CHIME) mission [16], which, unlike the previous hyperspectral remote sensing missions, is envisioned to observe continuously, as opposed to observing a handful regions of interest per orbit,

with a wider swath compared to the existing hyperspectral imagers. The envisioned continuous data acquisition rate of CHIME pushes the technical requirements of the mission. CHIME is proposed to feature a data downlink speed of 3.7 Gbit/s, which is more than an order of magnitude larger than the multispectral Sentinel-2 at 160 Mbit/s despite CHIME's smaller swath width. Because the CHIME hyperspectral instrument is proposed to gather data at excess of 5 Gbit/s, on-board cloud detection is necessary to fit the data within the downlink bandwidth [17].

On-board processing can be a solution for overcoming the downlink bottleneck for hyperspectral imaging satellites. Although it lacks both a hyperspectral imager and a specific scientific remote sensing mission, the OPS-SAT satellite is a demonstration of the potential for on-board processing in general [18], including image processing and machine learning models trained on-board the satellite after launch. The algorithms tested on OPS-Sat could in principle be adapted to run on larger satellite systems, such as CHIME. However, small satellites are often constrained by available processing power and energy [19–21]. Two recently launched cube satellites containing the HyperScout hyperspectral imager emphasize on-board processing capabilities. These are HyperScout on the GomX-4B satellite from 2018 [22,23] and HyperScout-2 on the FSSCat B/$^3$Cat 5B/Φ-Sat-1 satellite from 2020 [24]. They were designed to perform standard processing steps like georeferencing and radiometric calibrations on-board. In addition, HyperScout-2's ability for on-board cloud segmentation was demonstrated on the ground; however, there are no published results yet about the on-board data processing capabilities. The CHIME mission proposes several on-board processing techniques for various applications and has investigated two specific methods in depth by testing on simulated data [17]. The first is cloud detection integrated with adaptive compression methods based on CCSDS123 [25], adapting to the cloud content in an image. The second method is image segmentation based on Support Vector Machines (SVMs).

The authors of [26] briefly mention an issue with machine learning methods for hyperspectral remote sensing satellites, which is insufficient training data availability due to new sensors for each new mission. Their solution was to emulate the new camera sensor characteristics and generate simulated data. However, the data volume issue could also be solved via updates to the on-board software with new machine learning algorithms at a later stage during the mission, or via online learning [27]. Few previous missions had software updates as a part of the concept of operations to extend on-board processing functionality. Software updates of an operational nature have been performed during the operation of the Aalto-1 Spectral Imager mission [28]; however, no updates of on-board processing capabilities have been reported. Although it is yet to be launched, the Intuition-1 mission will develop a small satellite, a 6U CubeSat, to be a testbed for various Artificial Intelligence (AI) algorithms for hyperspectral data processing [29]. The previously mentioned OPS-SAT has received many updates to its software in the years since its launch [18,27].

Since cube-satellites do not require as large an investment as larger satellites, a few hyperspectral imaging cubesats motivated by specific applications have been launched. The GHG-sat constellation now consists of over 10 satellites that image with very fine spectral resolution within the short wave infrared portion of the spectrum in order to detect methane gas emissions [30]. While on-board real-time detection of methane has been demonstrated [31], the on-board payload processing is responsible only for downlinking data and telemetry and most of the data processing is still performed on the ground. The lack of on-board processing on GHG-sat, even though relevant algorithms have been demonstrated, suggests that additional research is needed to address concerns about its reliability or worth.

In contrast, the Hyper-Spectral Imager for Oceanographic Observations (HYPSO) mission, which uses visible/near-infrared imagers with a transmission grating design for algal bloom monitoring, includes on-board processing as a critical part of its mission [32–34]. A special focus of the first satellite in the constellation, HYPSO-1 (NORAD ID: 51053), launched on 13 January 2022, lies in reducing data and information latency from observa-

tion to availability to the end-user [35] because algal blooms are sporadic, spontaneous, and evolve continually. Therefore, a re-configurable on-board processing unit is included as part of the mission. Not only can HYPSO-1 deliver standard hyperspectral data products, it is also a research infrastructure for testing new on-board data processing algorithms. Similar to PROBA-1 in 2001, and recent enMAP and PRISMA satellites, HYPSO-1 can point off-nadir to facilitate timely imaging of chosen targets [32,36].

In this article, the on-board image processing pipeline of HYPSO-1 is presented. Various configurations of the pipeline achieve specific operational goals, which include the reduction of data volumes by lossless compression, mitigation of noise, extraction of features of interest, and computation of derived higher-level products [37]. The simplest of these variants, the Minimal On-board Image Processing Pipeline (MOBIP), has been executed over 1000 times in orbit. Fast processing and downlink enable early warnings for Harmful Algal Bloom (HAB) and the collaboration with other ocean remote sensing and in situ agents for real-time ocean monitoring. For example, if the spatial location of a possible algal bloom is imaged and identified by low latency on-board processing, that location information can be immediately downlinked and received by in situ agents, such as autonomous surface vehicles, which can travel to the location to investigate [35]. Because HYPSO-1 is driven by a specific, time-sensitive mission it shows clearly how an on-board image processing pipeline can be used to meet specific scientific and societal needs.

The scientific contribution of this paper is the design of a robust hyperspectral remote sensing satellite payload system with reconfigurable on-board processing capabilities. Its successful operation has been demonstrated on-board HYPSO-1 during its first year in orbit.

This paper is organized as follows. After providing a brief background in Section 2, the On-board Processing Unit (OPU) implementation, Minimal On-Board Image Processing (MOBIP) pipeline, and on-board processing pipeline architecture are described in Section 3. Testing and benchmarking of the system, as well as presenting in-orbit results are done in Section 4. Lastly, discussions and concluding remarks are provided in Section 5.

## 2. Background

### 2.1. Hyperspectral Imaging and Hyperspectral Data

Hyperspectral images from imaging spectrometers or hyperspectral cameras consist of $L \times S$ spatial pixels with $B$ bands each. Hence, a hyperspectral image is also referred to as a hyperspectral data cube with dimensions $L \times S \times B$. A hyperspectral data cube is usually captured by spatial or spatial–spectral scanning and usually contains many dozens and up to hundreds of bands, with bandwidths less than 15 nm. In a push-broom design like HYPSO-1, spatial scanning can be done via the satellite's motion, or by rotation of either the camera platform or an optical component.

There are three common kinds of data formats, also called sample ordering, that describe data streaming order from the sensor or data cube arrangement in memory: Band Interleaved by Pixel (BIP), Band Interleaved by Line (BIL) and Band Sequential (BSQ). The data serialization of a hyperspectral data cube can be understood as the mapping from the three-dimensional coordinates of the cube to a unique one-dimensional index. It is important to be aware of this format when implementing hyperspectral data processing algorithms and especially on Field Programmable Array Gate (FPGA), which accesses the raw data in memory directly. In the case of Hyperscout and its snapshot spectral camera, images are captured in BIL sample order and after georeferencing and radiometric calibration, the cubes are converted to BSQ, before gridding [22]. In a transmission grating hyperspectral camera, depending on sensor orientation with respect to slit, the raw data are streamed usually either in BIP or BIL format, with raw HYPSO-1 data using the BIP format.

### 2.2. Hyperspectral Data Processing

The amount of data per pixel in hyperspectral data can be an order of magnitude higher than multispectral data. In addition, with their higher dimensionality, hyperspectral

data are more difficult to visualize and extract desired information from than multispectral data and advanced data processing is required.

Advanced data-processing algorithms investigated by the HYPSO research team include the following:

- Band-ratio-based estimation [38],
- Spectral target detection [39],
- Dimensionality reduction [40,41],
- Deconvolution (deblurring) [42],
- Super-resolution by pan-sharpening [43,44],
- Super-resolution by slewing [32],
- and Compressive sensing [45].

On-board Hyperspectral (HS) data processing is constrained by large datasets, limited processing time and communication links. The Consultative Committee for Space Data Systems (CCSDS) has developed data and image compression algorithms CCSDS121 [46], CCSDS122 [47], CCSDS123 [25,48] specifically designed for space data systems, notably to reduce data transmission time [49]. The first issue of the CCSDS123 compression standard from 2012 [48] is an efficient lossless prediction-based algorithm characterized by low complexity. In fact, in recent years several FPGA implementations of the CCSDS123 standard have been presented in the literature [50–59]. The second issue of the CCSDS123 document [25] presents a lossy counterpart.

## 3. On-Board Processing Unit Design

The HYPSO-1 payload is developed in-house using mostly Commertial Off-The-Shelf (COTS) components and is integrated into a COTS satellite bus from Kongsberg NanoAvionics. The HYPSO-1 payload consists of three parts: the OPU, the Hyperspectal Imager (HSI) and a snapshot camera with Red Green Blue (RGB) channels. In addition, there is a wire harness for signal and power interfacing with the satellite bus. The HSI is a push broom design and consists of COTS camera and optical components, an in-house machined platform, and mounting hardware to the satellite frame via dampers [33,34,60]. The following subsections will present a detailed description of both the hardware and software of the OPU.

### 3.1. Design Goals

The space environment imposes a number of challenges on spaceborne electronic systems. Corrupted or non-functional software on a satellite can be caused by a faulty software update, by flipped bits in memory as a result of a radiation event, or by the degraded quality of the memory device. Among the most predominant risks for the payload system's function identified is in the boot process due to the criticality of data corruption. Improper booting may result in a critical failure causing the mission to end. Scenarios, where a reboot is required, include the reset of a non-responding sub-system or system power-off to save energy or to protect circuitry in the event of a solar flare. In the CubeSat mission Aalto-1, a total of 38 boot events occurred during the first five months of the mission, out of which 35 were unplanned [28]. In the mission AAUSAT-II, 11 spontaneous reboots every 24 h are reported [61]. This illustrates the importance of a reliable and fault-tolerant boot sequence.

Design Principles

The principles guiding the design of the OPU architecture are listed in Table 1. A description about how these principles were understood in the project follows.

**Table 1.** Overview of the guiding principles during the design and implementation of the OPU.

| Design Principle | Short Description |
| --- | --- |
| Integrity | Ability to detect faults |
| Redundancy | Presence of backups |
| Locality | Physical separation of backups |
| Modularity | System consisting of independent components |
| Extensibility | Ability to add additional functionality |
| Configurablility | Adjust functionality to user's needs |
| Low latency | Fast processing for fast available data |

The first three principles (integrity, redundancy, locality) are related to the startup or booting procedure, whereas the following four (modularity, extensibility, configurablility, low latency) are explored in applications for command handling, telemetry and on-board processing. *Integrity* is the ability to detect faults in the system, both in terms of faults due to corrupted data and in terms of software bugs. This is important as the space environment is known for latch-ups and single-event upsets that can cause system faults and data corruption. In addition, part of the concept of operation is to update the HYPSO-1 payload software frequently, and thus there is a risk of faults caused by software bugs. Verifying data integrity and detecting corrupted data can be achieved using hashes and checksums. *Redundancy* provides a backup solution in case of an integrity violation, to ensure continuous operation. *Locality* refers to how physically close in hardware backup copies of software or data are stored. Low locality reduces the likelihood that a fault event affects multiple copies. *Modularity* is a design property in which a system consists of independent components. High modularity reduces bugs, improves efficiency during development and is also beneficial for extensibility and simplicity. *Extensibility* refers to the ability to extend the system by adding new functions via updates. This is necessary as software development was not finished by launch, as well as for the ability to function as a research platform that allows to test new on-board processing algorithms as they become implemented and ready. *Configurablility* refers to the ability to adjust a function or task to meet specific needs by altering parameters without performing a software update. The operator or user may use different parameters to optimize information extraction depending on target of interest. Finally, *Low latency* is part of the mission concept of, e.g., HAB detection and communication with in situ agents for coordinated mapping and monitoring. These principles together aim for a high degree of fault tolerance. Additionally, the payload design was subject to constraints set by available time, knowledge, manpower and material resources.

### 3.2. Hardware Design

The OPU consists of a PicoZed System-on-Module (SoM), see Figure 1a, and an in-house designed carrier Printed Circuit Board (PCB) referred to as the Break-out Board (BoB), see Figure 1b. The PicoZed contains the Zynq XC7Z030-1 System-on-Chip (SoC) that has dual 667 MHz ARM Cortex-A9 cores and a Kintex-7 FPGA. The A9 cores of the SoC also feature the ARM NEON Single-Instruction-Multiple-Data (SIMD) extensions that are used to accelerate binning, see Section 3.4.1. The PicoZed was chosen due to flight heritage of the Zynq 7000 architecture [21,62,63] and because it features an FPGA which can be used to accelerate the heavy image processing tasks with sufficient performance to satisfy low latency requirements. Peripheral components of note are indicated in Figures 1a and 2. The BoB serves as an interstage between the electrical contact pins of the PicoZed and the wire harness interface to the satellite bus. It features an additional 8 GB SD-Card and has connectors for data and power interfaces shown in Figures 1b and 2. The OPU fully assembled can be seen in Figure 3a in the middle right, between two aluminum panels.

The Hyperspectal Imaging (HSI) camera utilizes the monochrome version of the Sony IMX249 camera sensor inside the COTS camera module UI-5261SE Rev. 4 from IDS Imaging Development Systems GmbH [64]. It has a pixel array that is 1936 pixels wide by 1216

pixels high. When integrated with the optical components, the camera forms a transmission grating push broom camera where light is dispersed along the width of the pixel array and the sensor height corresponds to the spatial, along-slit dimension. The Sony IMX249 sensor features 12-bit sample depth and the camera module has a maximum frame rate of 47 fps at 8-bit and 23 fps at 12-bit over its Ethernet interface that has a rated transfer speed of up to 1 Gbit/s [64]. The camera features electrical Input/Output (I/O) pins for capture triggering and flash signaling. The capture trigger pin is not used. The camera manufacturer provides a driver and C Application Programming Interface (API) for recording data.

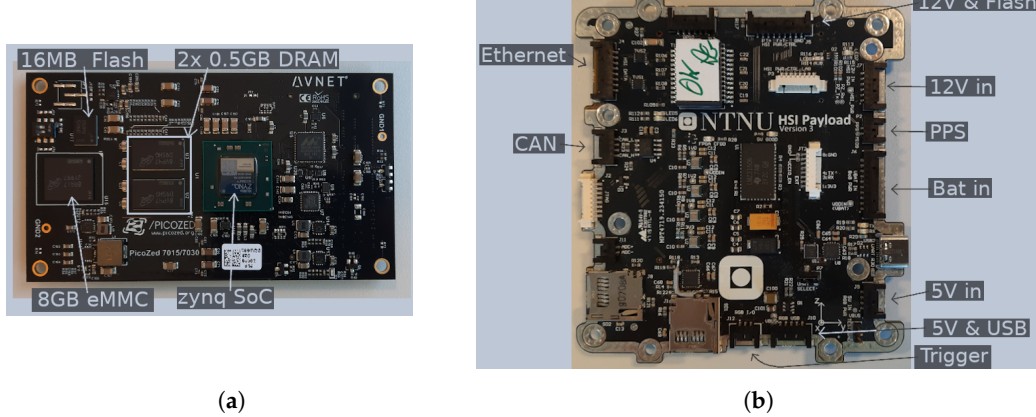

(**a**) (**b**)

**Figure 1.** Images of the PicoZed SoM (**a**) and the BoB (**b**). Connectors for mounting the two together are on the other side not shown here. Components in Figure 2 indicated.

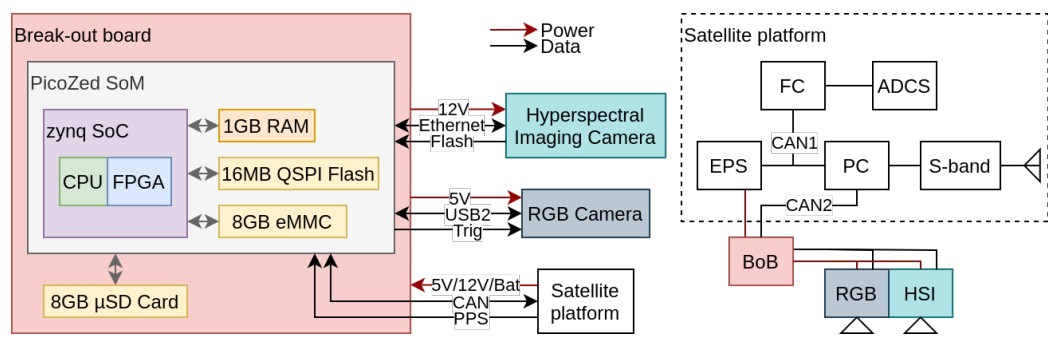

**Figure 2.** Overview of the most important hardware components in the payload and on the satellite bus and their interfaces. This figure shows a simplified payload-centric view to the left, and a simplified satellite-bus-centric view to the right.

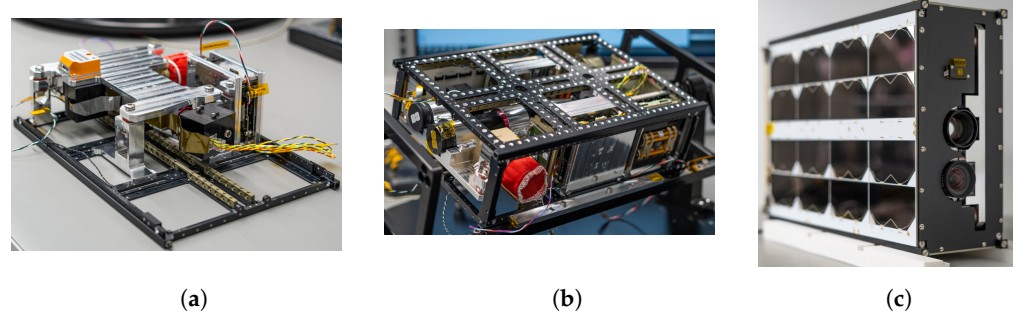

(**a**) (**b**) (**c**)

**Figure 3.** Images of HYPSO-1 during assembly, integration and testing. (**a**) Payload components. (**b**) Full satellite without panels. (**c**) Full satellite assembled.

The payload is integrated into a COTS satellite bus. Notable satellite bus subsystems are the Attitude Determination and Control System (ADCS), the Electronic Power System

(EPS), the Flight Computer (FC) and the Payload Controller (PC), and the S-band radio. The ADCS subsystem, which is configured via the FC, controls the orientation of the satellite. The Pulse per second (PPS) from the Global Navigational Satellite System (GNSS) receiver in the ADCS is used for precise time synchronization. Main data, telemetry and command communication with ground stations are done via the S-band radio. Both the FC and the PC contain script engines to schedule actions outside of ground station contact. The PC also has a high-speed connection to the S-band radio for faster downlink than from any other subsystem via the Controller Area Network (CAN) bus, and contains storage to which data can be buffered to from other subsystems. The power to all subsystems including the payload is controlled by the EPS.

### 3.3. Software Design

This subsection is organized in three parts, starting with a description of the booting procedure and operating system in Section 3.3.1, and what choices were made in their design with regard to integrity, redundancy, and locality. The second part discusses the main application of the OPU in Sections 3.3.2 and 3.3.3, and the third part discusses the on-board processing pipeline application and architecture in Section 3.4.

### 3.3.1. Booting Procedure and Operating System

Following the power-on event, the OPU boots the embedded Linux Operating System (OS). After booting, the SD-Card and eMMC are mounted, the camera drivers are loaded, time is synchronized between the OPU and satellite bus, and lastly, the main application is executed which is responsible for command and data handling. One essential feature in the design of the HYPSO-1 OPU is that the payload is expected to be powered-off most of the time, and is only powered-on for operations specifically involving communication with the OPU payload subsystem such as image acquisition or data buffering [32]. This conserves energy and the chance for single-event upsets is reduced. Two software components are involved in booting: the bootloader and the Operating System (OS) image.

The bootloader consists of a First Stage Bootloader (FSBL) and a Second Stage Bootloader (SSBL). The OS is a customized embedded Linux distribution that is built and packaged into a boot image using the Petalinux toolchain provided by the designer of the Zynq XC7Z030-1 SoC AMD Xilinx. A booting procedure with integrity and redundancy in mind is designed as shown in Figure 4, by using the scripting capabilities of the U-boot SSBL [65]. Both the FSBL and the SSBL are stored thrice in sequence on the Quad Serial Peripheral Interface (QSPI) Flash. The detection of potential data corruption in the SSBL is ensured by md5 checksum. If verification by checksum fails, booting will restart using the next copy of the SSBL on the QSPI Flash.

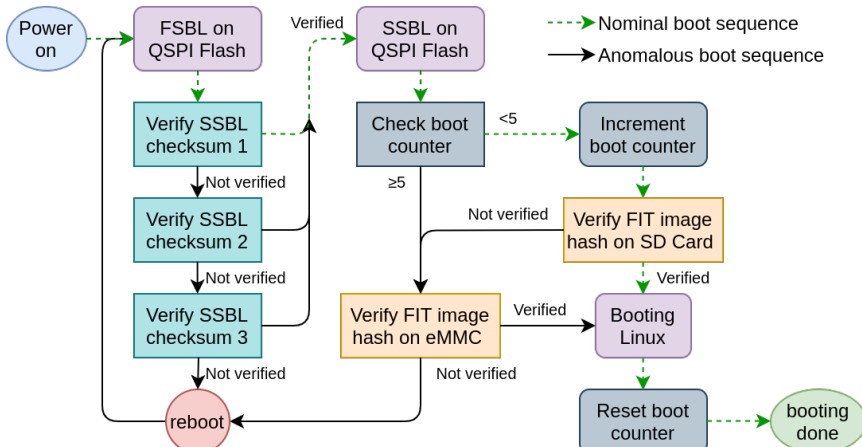

**Figure 4.** Flow chart of the boot sequence on the OPU which highlights the sequence of integrity checks and use of backups during the booting. Nominal booting without any integrity errors is indicated by the green dashed arrows. If the *reboot* node is reached, the OPU is unrecoverable.

The OS image file in Flattened Image Tree (FIT) format contains all components needed for booting Linux. There are two FIT boot image files on HYPSO-1. The primary image stored on the SD-Card, and the backup image, also named the golden image, stored on the embedded Multi Media Card (eMMC). The image on the SD-Card is the active configuration that is booted during fault-free operations. See Section 3.5 for a description about updates to the on-board software. The SSBL verifies data integrity of the boot image file using a hash and creates a root file system in volatile Dynamic Random Access Memory (DRAM). This has the advantage that no critical system files are stored outside of the hash verified image. All files to be kept between power cycles are stored on the SD card or eMMC outside the boot image file. These are non-system critical files like HSI data. There is no persistent OS state that can be corrupted permanently without being detected. If the data integrity verification fails, the SSBL attempts to boot the golden image on the eMMC. Once the golden image is booted, the corrupted boot image file can be replaced with the golden image file or an uplink of a non-corrupted image file can be performed. The SSBL is configured to increment a counter at the start of its execution that is also stored on the QSPI Flash. This counter, named boot counter, is reset on successful boot of a boot image file. If a bug in the SD Card image causes the boot process to not finish, the boot counter is not reset. After the counter reaches the value 5, the SSBL boots the golden image instead, without checking the primary image on the SD Card. In addition, using the boot counter, booting of the golden image can be forced in cases where booting is successful, but due to a software bug no communication can be established. This is done by power cycling the OPU five times before booting is completed and the counter is being reset. The faulty boot image file can then be corrected from the golden image.

In addition to basic system files, the boot image file also contains the main application software as well as the libraries and drivers that it requires. The main application software is responsible for command handling, HSI control, file transfer and telemetry, as well as on-board processing pipeline configuration and control. The application software comes as a single application file, which is run as the last step in the boot sequence. Before the main application is run, time is synchronized between the OPU and the satellite platform, as the OPU does not feature an on-board Real Time Clock (RTC). The OPU system time is set by requesting the time of the RTC on the FC, which is in turn set using the on-board GNSS module of the satellite bus.

### 3.3.2. Application Software and Payload Control

The payload control software running on the HYPSO-1 OPU has been described in [66]. In short, at the end of booting, the main application software is started by a startup script. The main application software consists of multiple parallel running threads that implement services. A complete list of service threads is shown in Table 2. The majority of the threads listen for incoming commands via the Cubesat Space Protocol (CSP) [67], where the commands are predefined in code and may contain parameters. The HYPSO team has developed a utility to send these messages from a ground station to the application software using a human readable Command Line Interface (CLI). The CLI and the main application software use a modified version 1.5 of CSP. These messages trigger various actions such as initiation of HSI data capture, file transfer and data buffering. The satellite bus, specifically the Flight Computer (FC) and Payload Controller (PC) subsystems, feature script engines, with which CSP packets with arbitrary data can be scheduled to be sent to any satellite subsystem at specified points in time without ground station contact. In addition to sending arbitrary CSP packets to the OPU, the scripts on the FC control satellite pointing or slewing, and power control to the OPU and HSI. After capture, either immediately or with some delay, HSI images are buffered to the PC outside Ground Station (GS) passes via the slow CAN interface. By doing so, downlink speed improves during a GS pass due to a fast direct communication link between the PC and the S-band radio.

**Table 2.** List of service threads started as part of the application software.

| Name | Long Name | Description |
|---|---|---|
| CSP | Cubesat Space Protocol | Services included in the CSP library mainly used for pings |
| FT | File Transfer | Handles file down- and up-load requests |
| CLI | Command Line Interface | Provides OS shell access |
| RGB | Red–Green–Blue Camera | Handles commands to control the RGB camera |
| OBIP | On-Board Image Processing | Runs on-board processing pipeline tasks |
| HSI | Hyperspectral Imager | Handles commands to control the HSI camera |
| TM | Telemetry | Collects and logs telemetry information |
| LOG | Logging | Handles information and error message logging. Creates log files. |
| Monitor | Service Monitor | Responsible for start and stop of the above tasks |

### 3.3.3. Hyperspectral Data Capture

In the idle state, the OPU is powered off. A few minutes before a pass over a capture target, the OPU is powered on via a PC script, satellite pointing is configured via a FC script, and capture is initiated via command to the HSI service thread. The HSI service thread uses the uEye driver C API from iDS imaging for camera control [68]. See Table 3 for a list of some capture-influencing parameters that can be configured as part of a data capture command. The Sizing parameters influence the resulting hyperspectral cube dimensions, which are $L \times S \times \lceil W/(f_b f_s) \rceil$. The line count and frame rate parameters determine the total capture duration, which is $L/f_r$. Exposure time is usually set within a range of 10–40 ms, but can be set to any value from 0.02 ms up to what the configured frame rate allows. The gain is usually kept at the lowest setting for the best possible Signal-to-Noise Ratio (SNR). The sizing, timing and signal level parameters influence spatial resolution, spectral resolution and SNR as discussed in [32,69].

**Table 3.** List of parameters configurable via capture command. Some parameters, e.g., for debugging are omitted.

| Group | Parameter | Description |
|---|---|---|
| Sizing | Line count, $L$ | How many lines to scan. In other words, how many frames to capture. |
| | Sensor height, $S$ | Spatial sampling. Default are 684 pixels. |
| | Sensor width, $W$ | Spectral sampling. Default are 1080 pixels. |
| | Binning factor, $f_b$ | Software binning in the spectral dimension. Possible values: $1\times$ though $18\times$ |
| | Sub-sampling, $f_s$ | On-sensor subsampling in the spectral dimension. Possible values: $2\times, 4\times$ |
| Timing and signal level | Frame rate $f_r$ | Rate at which a line/frame is scanned/captured. |
| | Exposure time, $e$ | How long light is collected during a line/frame scan/capture. |
| | Gain, $g$ | |
| Flags | CCSDS123 in software | Compress HSI data using the software implementation of CCSDS123 instead of the FPGA implementation |
| | No compression | Do not compress the HSI data and store the uncompressed data on SD Card instead. |

The camera is configured to output digital flash signals for every frame. These flash signals are captured and timestamped by the time-stamping module on the FPGA. The

image data from the camera are fetched by the driver and stored in a frame buffer queue in DRAM. The frame buffer queue is dynamically allocated with the size of each frame buffer is computed as $2 \times S \times W$ Bytes, with the factor 2 corresponding to the sample size of 2 Bytes per sample.

### 3.4. On-Board Processing Pipeline

Processing pipeline refers to the software–hardware co-design for capturing and processing hyperspectral data consisting of a predefined set of processing stages. At launch only a minimal set of on-board processing features were implemented. This minimal set of processing features, the Minimal On-Board Image Processing (MOBIP) pipeline, is shown in Figure 5. Many other processing features are in development, as well as an application implementing a reconfigurable on-board processing pipeline framework which is separate from the main application software, see Figure 6. This framework is described in Section 3.4.2.

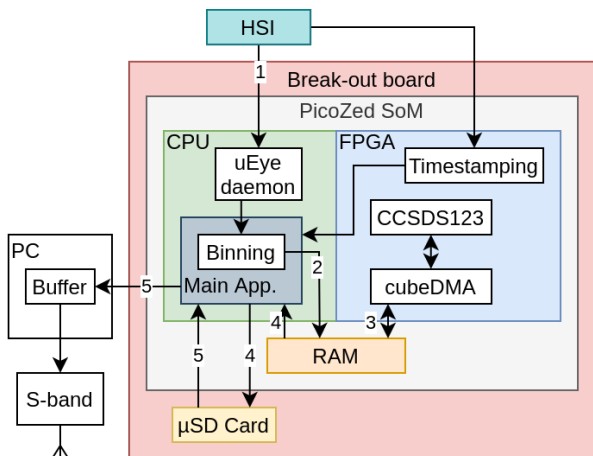

**Figure 5.** Illustration of the data flow across the subsystems and components of the HYPSO-1 satellite in the MOBIP pipeline. 1. Low-level camera control and data transfer by the ueye driver. 2. Online binning of individual lines/frames and storage of the binned cube data into a buffer in main memory. 3. Transfer via cubeDirect Memory Access (DMA) and CCSDS123 compression on FPGA. 4. Storage of compressed cube on SD Card. 5. After some time, buffering followed by downlink.

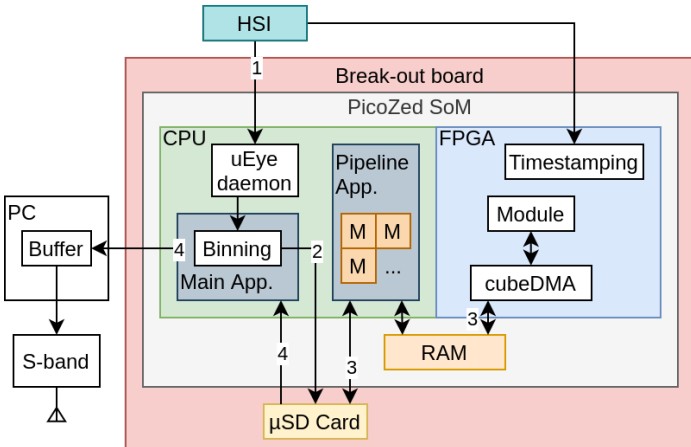

**Figure 6.** Illustration of the data flow across the subsystems and components of the HYPSO-1 satellite using the general pipeline framework. Differences to Figure 5 are: Storage of the online binned cube data to SD Card (2) and Loading of binned cube data to main memory for processing in software or on FPGA (3). The arrow from Timestamping to Main App. is omitted in this figure, for simplicity.

### 3.4.1. Minimal On-Board Image Processing Pipeline

The MOBIP pipeline consists of setup, HSI data capture (see Section 3.3.3), binning, compression, and post steps. The setup and post steps include the following low-level processing steps:

- parsing capture command packet,
- allocating memory,
- starting and stopping the time-stamping module,
- opening the camera and setting parameters,
- starting and stopping recording,
- and creating log files and metadata files.

Data binning is a pre-processing technique where the original samples in small intervals (bins) are replaced by a value representative of that interval. In the context of image processing, binning combines a cluster of pixel samples into a single pixel sample. In the context of hyperspectral imaging, both spatial and spectral binning can be applied. Although associated with loss of information, binning reduces the amount of data to be processed facilitating further analysis and decreasing challenges related to data storage. Binning may also reduce the impact of noise on the processed image (i.e., increased SNR) at the cost of a lower resolution. Among different binning methods, median- and mean-based binning methods have been considered. The current pipeline implementation employs mean binning in the spectral dimension, but without division by sample count. The data type in which the raw hyperspectral data sample values are stored is 16-bit unsigned integer, despite the sample depth of the sensor being 12-bit, and it is possible to add 12-bit values up to 16 times without risking overflow.

Due to the theoretical spectral bandwidth of the optical system of around 3.33 nm, the high raw spectral sampling of 0.37 nm [32], and the limited available storage in DRAM, the default binning factor is set to $9\times$. Online binning during capture is required to fit enough lines for sufficient area coverage into main memory. This imposes a processing time constraint, requiring that binning on an image frame is done before the next frame is exposed and transferred to the SoC. Hence, mean binning is chosen as median binning requires more processing time. The binning factor can be configured on a per-capture basis to any value between $1\times$ and $18\times$. In [36], it is shown that a lower binning factor can give higher spectral resolution despite sampling below the theoretical bandwidth, which could potentially aid in spectral target detection. Increasing binning factor where spectral resolution is not a priority can be done to increase the SNR. This is important when imaging targets with little reflectance. Binning factors $4\times$ to $18\times$ are implemented using the ARM NEON SIMD instructions.

After image capture and binning, with a full binned cube in main memory, a DMA Intellectual Property Core (IP Core) has been developed, cube DMA [70], specialized to efficiently stream hyperspectral data to the FPGA processing modules in the sample order required by the FPGA module irrespective of whether the HSI data are stored in BIP, BIL, BSQ in the main memory.

A fraction of the on-board DRAM, typically 0.5 GB or 0.75 GB, is reserved for HSI data for transfer to and from the FPGA using cubeDMA. An API for interfacing with cubeDMA and transferring data is defined via a Linux kernel module writing to control and configuration registers on the FPGA.

### 3.4.2. Reconfigurable Pipeline Framework

A more general setup shown in Figure 6 reveals two main changes and additions: Another application for on-board software processing, and a framework for reconfiguring the FPGA with processing modules other than CCSDS123 compression. The MOBIP pipeline will be one specific configuration of the general pipeline framework.

The architecture of the on-board processing pipeline application is implemented as shown in Figure 7. The pipeline is a single C program, consisting of the pipeline engine and the pipeline modules. Each module is configured with its own module configuration

file. A pipeline configuration file defines which modules to run as a pipeline stage and the order in which they are run. Stages need not depend on each other. A stage may or may not interface with the FPGA or reconfigure the FPGA. Different sets of configuration files are stored on-board for different pipelines. Different pipelines are run depending on the imaging target.

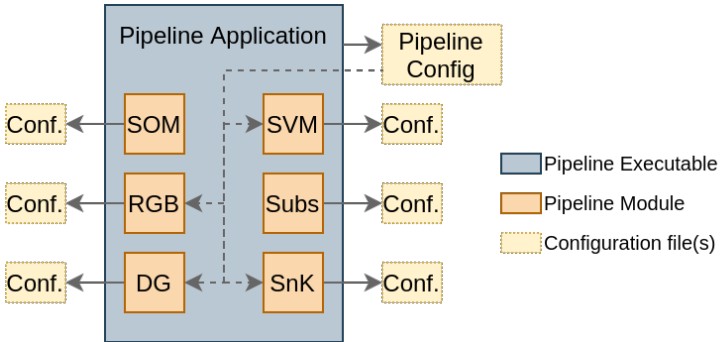

**Figure 7.** Architecture of the on-board processing pipeline. The six modules are the modules currently implemented and tested on the ground. See Table 4 for a description of the stages. In this example, indicated by the dashed arrows, the stages SVM, RGB and DG and SnK are picked to be run via the pipeline configuration file.

The setup and capture stages of the MOBIP pipeline are (up to capture parameters) included in each pipeline configuration, as recorded hyperspectral data will always be required. The same holds for time stamping and binning.

The workflow for additional pipeline stages consists of algorithm prototyping in Python, implementation and testing in C, potential parallel implementation in Hardware Description Language (HDL) such as VHSIC Hardware Description Language (VHDL) or in a High-Level Synthesis (HLS) language and testing on Programmable Logic (PL), and lastly integration and testing on target hardware before updating the software on HYPSO-1. Currently implemented stages and their motivation are shown in Table 4. These stages have been tested on the ground [71].

**Table 4.** Smile and Keystone (SnK) correction: camera calibration. Direct Georeferencing (DG): on-board estimation of coordinates of the spatial pixel.Self-Organizing Map (SOM): dimensionality reduction. Support Vector Machine (SVM): segmentation. Subsampling (Subs.): extract a subset of spatial pixels. Red Green Blue (RGB): Composition of an image using three bands from the datacube.

| Abbreviation | Motivation |
|---|---|
| SnK | Improve data quality |
| DG | Pass target information to in situ agents |
| SOM | Package data for quick download |
| SVM | On-board data segmentation |
| Subs | If only parts of a cube are interesting |
| RGB | Preview a capture |

### 3.5. Software Updates

Updates to the software of HYPSO-1 can be of two types: application software updates and firmware updates. Application software updates are newer versions of the main application or the pipeline application file stored on the SD Card or eMMC. Firmware updates are updates of the primary boot image on the SD Card and newer versions of the FPGA image file or additional FPGA modules. At the launch of HYPSO-1, the golden image and the primary image were identical. During nominal operations, the primary image is expected to be updated while the golden image is to be kept at its initial launch version. The motivation behind distinguishing application software and firmware updates

is primarily due to their size and the level of risk associated with performing the updates. The size of a boot image is ca. 17 MB and the size of an application file is 0.5–1 MB. Updates will likely increase these sizes, thereby increasing operational complexity as the upload rate is limited. A newer version of the main application identified by a file path and stored on the SD Card can be run on command. Similarly, the pipeline application is run by command and the version is identified by path. Thus, multiple versions of both the main application and pipeline application can be stored on-board and run as required. See Table 5 for an overview of files to be stored on the permanent storage media on the OPU. Experimental newer versions, that may contain bugs, are run as required. A safe version of the main application that is stored in the hash-verified boot image is run after power-on and boot. This enables error recovery via reboot when running newer versions.

**Table 5.** Overview of the most important files and kinds of files stored, and intended to be stored on permanent storage available on the OPU. *Italics* means files uploaded or generated post-launch.

| SD Card | eMMC |
| --- | --- |
| Boot image | Boot image (golden) |
| FPGA image | FPGA image (golden) |
| *HSI capture data* | |
| *log and telemetry files* | |
| *updated main application files* | |
| *updated FPGA modules* | |
| *updated pipeline application files* | |
| *pipeline configuration files* | |

## 4. Testing and In-Orbit Results

This section presents testing results after software development was largely finished. These tests are used to determine the valid ranges and combinations of the capture parameters shown in Table 3. In-orbit results are presented in Section 4.4. Details on software-testing strategies, materials and lessons learned were presented in [72].

### 4.1. Firmware Integrity Tests

The three integrity checks during boot shown in Figure 4 are independently verified.

The triple redundancy of the SSBL image file is verified by sequentially corrupting each of the three SSBL images by erasing 10-byte chunks of the file on the QSPI Flash and attempting to boot. The results are as follows:

- Test 1: No data corrupted: System boots correctly.
- Test 2: First copy of SSBL corrupted: System boots correctly.
- Test 3: First and second copy of SSBL corrupted: System boots correctly.
- Test 4: All three SSBL corrupted: System fails to boot.

The tests show that the checksum mechanism is able to detect the corrupted bootloader and proceed to load the backup until all are corrupted.

The test to verify automatic fallback to the golden image is done by simulating bitflip in the primary image file and by ejecting the SD card to simulate a hardware issue. In addition, boot counter overflow is simulated by scripting five power cycles of the payload with a period shorter than the boot time. On the sixth power-on event, the OPU is left to boot fully. The results are as follows:

- Test 1: No data corrupted: Primary image boots correctly
- Test 2: SD card not present: Golden image boots correctly
- Test 3: Primary image is corrupted: Golden image boots correctly
- Test 4: Five power cycles: Golden image boots correctly

For development and debugging, the boot image is configured to include Dynamic Host Configuration Protocol (DHCP) and an Secure Shell (SSH) server. For deployment, these are disabled. With them, booting takes 37 s, and the application software is not

started automatically. Instead, a serial terminal is available for login and command line shell access. Booting a deployment image takes 19 s due to the omission of DHCP discovery and generation of the Rivest–Shamir–Adleman (RSA) key pair for SSH.

### 4.2. Ethernet and Data Recording

It was noticed that the Central Processing Unit (CPU) of the chosen SoC does not have sufficient processing power to record at the maximum advertised frame rates of the camera module.

A number of factors made predicting the achievable frame rate of the camera system difficult, among these are limited CPU performance with respect to processing power and Input/Output (I/O) bandwidth, Linux kernel configuration or various system implementation choices. The proprietary driver for the COTS camera module was also a source of uncertainty. Knowing the achievable frame rate is important from an operational satellite perspective because it influences the ground resolution, the SNR and total recording duration. Thus, throughput and data-recording tests were needed to characterize the capability of the system.

For throughput testing, nuttcp tool was used. The HSI camera was disconnected and replaced by a desktop computer. A single nuttcp test works by transmitting data packets from the desktop computer to an nuttcp server on the OPU at a specified constant data rate for a specified duration of time. nuttcp tracks how many packets are received and how many are dropped and reports the results. Many single tests were repeated while varying the transmitting data rate. Three test sets exemplary of the Ethernet performance are shown in Figure 8, consisting of one sparse test varying transmission rate from 140 Mbit/s to 500 Mbit/s in 20 Mbit/s steps and two dense tests with smaller step sizes. This testing found throughput to drop in reliability at around 260 Mbit/s, which is about a quarter of the theoretical maximum speed of the physical link. This means to achieve the maximum advertised frame rate, the number of pixels that is being read out from the sensor must be reduced to a quarter. This motivates the choice of nominal dimensions 1080 by 684 pixels as described in [32].

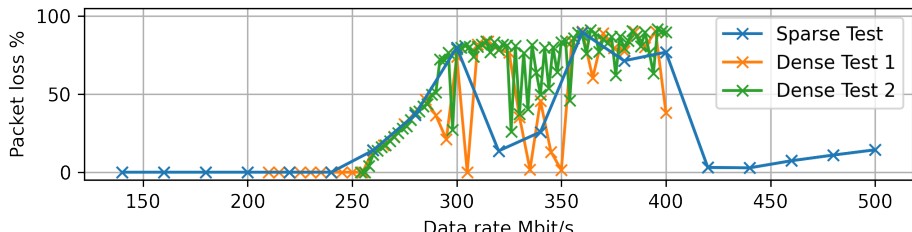

**Figure 8.** Figure showing packet loss percentage at various data rates.

Testing the frame rate capacity of the system was done by storing software timing information during capture. By comparing this to hardware timing information from the time-stamping module, it was determined how many frames were dropped, indicating that a bottleneck is limiting Ethernet throughput. Additional frame rate testing was necessary because the throughput tests alone were not a reliable predictor of frame rate capacity at different configurations.

As Linux is not a real-time operating system, data recording performance is inconsistent. An accurate picture was achieved by experiment repetition and averaging over many samples. Individual capture tests were repeated to determine performance statistics. Capture repetitions were performed in sets of 5. Each set had a given frame rate and resolution. The OPU was rebooted between each set. Figure 9 shows the data gathered from an example run of three capture sets at, respectively, frame rates 23 Hz, 24 Hz and 25 Hz. The plots show the time between frames measured in software. All captures were at nominal resolution of 684 samples by 1080 bands and recording 500 frames. The ideal plot in a system without bottlenecks would show perfectly regular times, meaning the time

between frames is very close to $\frac{1}{f_r}$. This is almost the case in the top plot of Figure 9 when recording with a frame rate of 23 Hz. This capture set shows almost consistent values of around 43.5 ms, with one anomaly in one recording near frames 440–450. A further test sequence was done over 30 captures for each of 5 different frame rates from 21 Hz to 25 Hz. The number of captures for each frame rate contains dropped frames is shown in Table 6. For safe overhead, the maximum nominal frame rate of 22 Hz was chosen, even though 23 Hz did not show any captures with dropped frames during testing. This method was repeated for more sensor resolution and binning modes to determine their nominal frame rates shown in Table 7.

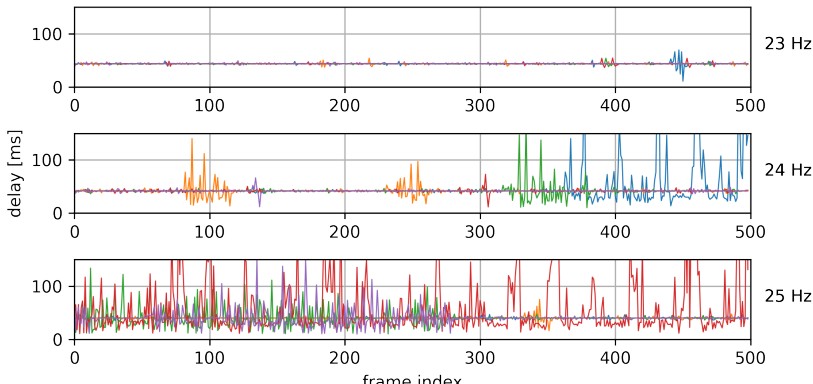

**Figure 9.** Plots of the time interval between when the uEye daemon reports new frames to the main application (i.e., the delay between each frame). Colors indicate the capture repetition, where blue, orange, green, red and purple are respectively the first, second third, fourth and fifth capture in the capture set. Above a frame rate threshold, timings become very irregular, and dropped frames become a possibility.

**Table 6.** Testing recording consistency. A bad recording is one that contains dropped frames.

| $f_r$ [Hz] | Bad Recordings/Total |
|:---:|:---:|
| 21 | 0/30 |
| 22 | 0/30 |
| 23 | 0/30 |
| 24 | 3/30 |
| 25 | 29/30 |

**Table 7.** The maximum nominal frame rate at different resolution and binning capture modes. Sub 2× means on-sensor sub-sampling in the spectral dimension.

| $f_r$ [Hz]/Mode | 1936 × 1216 | 1080 × 1194 | 1080 × 684 |
|:---:|:---:|:---:|:---:|
| SIMD Bin9 | 5 | 12 | 22 |
| SIMD Bin18 | 5 | 12 | 22 |
| SIMD Bin9 Sub 2× | 9 | 20 | 36 |

### 4.3. Compression

The cubeDMA and CCSDS123 IP cores have been verified by streaming data of size 512 × 2000 × 128 collected by HICO imager [7] for a variety of generic parameter cubeDMA and CCSDS123 values. The resulting processed data streamed by cubeDMA to and from the CCSDS123 core back to the memory have been successfully compared to the compressed data generated by the CCSDS123 reference software Emporda [73].

The time taken for CCSDS123 compression is necessary to know for scheduling tasks on-board. Figure 10 shows the test results of compressing data cubes with varying dimensions. The compression time did not vary depending on data content, only depending

on data size. Notably, the nominal resolution of 956 × 684 × 120 takes about 398 s in software and 186 ms on FPGA. This corresponds to a decrease in throughput compared to the results in [58] from 750 MSa/s to 422 MSa/s. This is explained by overhead that is present when integrating an IP Core into a complete embedded system with a non real-time Linux OS. Further FPGA compression tests for different cube dimensions were performed by compiling multiple FPGA images corresponding to different cube dimensions, booting up the OPU, and uploading the FPGA images to the SD Card. The FPGA was reconfigured using the CLI service of the main application. The following empirical formula for software compression time is determined by observing the linear dependence on cube dimensions and fitting a likely expression to the test data in Figure 10a,

$$t_{\text{sw}} = 5.1 \times L \times S \times B$$

where $t_{\text{sw}}$ is the time required for software compression in microseconds [µs], and $L$, $S$ and $B$, respectively, refer to the lines, samples and bands of the data cube. This expression is successfully used during operation to schedule actions on the satellite when not using recording with nominal cube dimensions where fast FPGA compression is not currently possible.

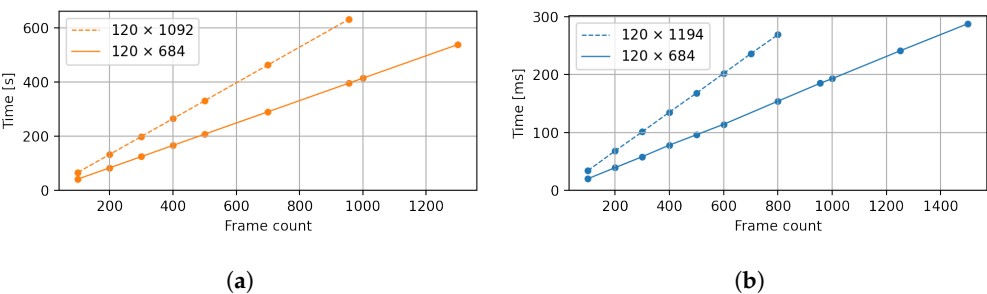

**Figure 10.** Length of time taken by long compression in software and on FPGA on the OPU. (**a**) Software compression. (**b**) Compression on FPGA.

### 4.4. In-Orbit Results

HYPSO-1 was launched on 13 January 2022 aboard a Falcon 9 rocket flying the SpaceX Transporter-3 Mission. Since then, as of January 2023, more than 1000 hyperspectral data cubes have been recorded and downlinked [36], which is more than 188 GB of hyperspectral data. With the average compressed datacube file size being about 70 MB, this corresponds to about 8 days of downlink time over the 1 Mbit/s link of the S-Band radio. Only the ground station at NTNU was used initially. Later on, the KSat ground stations on Svalbard via the KSATlite service were used for data downlink as well. However, because the passes over KSATlite on Svalbard and over NTNU overlap, this did not result in a big improvement in downlink capacity. During operations, it was found that over the course of a day, a single ground station is able to downlink about 5–6 images with nominal dimensions. Operations are ongoing and targets are imaged daily.

Corresponding to the 1000+ captures, the OPU performed more than 1000 boot sequences with at least 300 additional boot sequences for delayed buffering and manual operation. The second SD card slot shown in Figure 1b was intended to store another boot image for triple redundancy. This could not be implemented due to a technical design error. However, the OPU is yet to boot the golden image on the eMMC, indicating that the degree to which integrity and redundancy was included in the system was adequate.

Considering the 80 most recent captures before 19 April 2023, the average delay until the whole data cube was downloaded was about 432 min, which is 7.2 h. Frame rate performance, binning timing, software and FPGA compression timing are consistent with on-ground testing. The nominal cube dimensions were used most. However, for some captures other cube dimensions were configured, including, for example, for the purpose

of wider spatial coverage or calibration purposes. See Table 8 for a distribution out of a selection of 1043 captures.

A software update of the main application was performed by uploading a newer version of the program to the SD card and running this version from a PC script after the OPU has booted. This was a successful test, although the newer version did not add substantial new functionality.

**Table 8.** How often the four most configured cube dimensions were used out of a selection of 1043 captures.

| Configuration Name | Dimensions $L \times S \times B$ | Count |
|---|---|---|
| Nominal | $956 \times 684 \times 120$ | 895 |
| No binning | $106 \times 684 \times 1080$ | 34 |
| Full sensor | $33 \times 1216 \times 1936$ | 26 |
| Wider spatial | $598 \times 1092 \times 120$ or $537 \times 1216 \times 120$ | 88 |

Energy Usage

The energy use of the first HYPerspectral Smallsat for Ocean observation (HYPSO-1) payload measured by the EPS is plotted in Figure 11. The energy use of the payload is shown during recording, processing and buffering of an image. This operation consists of four phases. These are:

1.  Preparations for image recording (seconds 45 to 165).
2.  Image recording (seconds 165 to 210).
3.  Post actions, including software compression (seconds 210 to 730).
4.  Buffering the data to the PC (seconds 1990 to 4408).

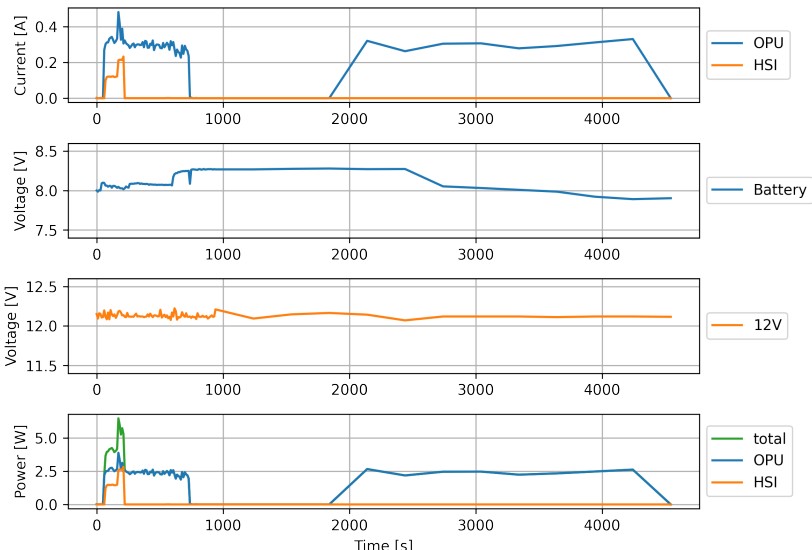

**Figure 11.** Plot of energy-use-related telemetry of the OPU and HSI from the EPS. Top to bottom: Current draw of the OPU and the HSI, Battery voltage to the OPU, 12V to the HSI, power use of the OPU and the HSI.

The HSI camera is turned on only for phase 1 and 2. The EPS telemetry logging rate was increased during phases 1 to 3, but not during phase 4. The telemetry logging rates were 10 s and 5 min, respectively. This image was recorded using a wider spatial imaging mode, for which compression on FPGA is not implemented and software compression is used. Software compression makes phase 3 take about 520 s. The OPU is directly connected to the EPS battery which has a voltage of around 8 V. The HSI camera is connected to a 12 V output of the EPS. Power draw of the payload peaks at just above 6 W during imaging.

See Table 9 for mean power draw and total energy used during the image acquisition. The OPU is powered on 77.93% of the time transferring data to the PC, corresponding to 75.1% of the total energy use. Note the battery voltage barely drains during imaging, and slightly jumps up during phase 3. This is due to the satellite orienting itself for solar charging again after pointing towards an imaging location.

**Table 9.** Overview of the mean power draw and energy use of the OPU and HSI camera during a single image-recording operation. Percentages indicate fraction of the total.

| Phase | Duration [s] | OPU [W] | HSI [W] | Energy OPU & HSI [Wh] |
|---|---|---|---|---|
| 1 | 120.0 (3.87%) | 2.35 | 1.20 | 0.12 (5.58%) |
| 2 | 45.0 (1.45%) | 3.09 | 2.65 | 0.08 (3.70%) |
| 3 | 520.0 (16.76%) | 2.35 | 0.00 | 0.34 (15.61%) |
| 4 | 2418.0 (77.93%) | 2.43 | 0.00 | 1.62 (75.10%) |
|  |  |  |  | Total energy: 2.16 Wh |

## 5. Discussion, Future Work and Conclusions

The design principles after which HYPSO-1 was designed were presented in Section 3.1. On-ground tests showed the resilience against faults via integrity and redundancy; however, there has not yet been an in-orbit event putting this into action. While the system is designed to be redundant, there is still one design flaw through which a single-bit error can break the OPU, which is in the FSBL. This is work in progress and will be addressed in a future publication.

As discussed in [32], further on-board processing algorithms are planned and will be uploaded to the satellite and tested. The next ones are RGB composite/thumbnail generation, georeferencing and segmentation using SVMs and binary decision trees. As HYPSO-1 continues to generate data, the potential for the successful training of machine learning models and implementation for on-board processing on HYPSO-1 is increased.

Sensor calibration represents much of the work undertaken by the team since launch [36], in addition to identifying methods to succesfully optimize, automate and schedule the operation of HYPSO-1. As we move on to the next phase in the mission and integrate software updates and reconfigurable on-board processing pipelines in the operation, similar work on learning how to operate the satellite optimally is likely required. A future update to the main application will include on-sensor subsampling in the spatial dimension of the sensor (see Table 3) to enable higher frame rates at a sensor resolution of $1080 \times 1194$ or similar at the expense of across track spatial resolution, but without losing spectral resolution.

The average number of daily images can not exceed about 5–6 images. Otherwise, there would be more images recorded than there is capacity to downlink with a single ground station. Thus, the energy use per orbit varies depending on whether an image was recorded during that orbit. However, it is possible to determine the average energy use per orbit. The payload uses about the same amount of energy as when recording a single image. When assuming 5.5 images per day, 2.16 Wh per image recording (Table 9) corresponds to an average daily energy use of 11.88 Wh. With about 15 orbits per day, the HYPSO-1 payload uses about 0.79 Wh per orbit. Comparing the in-orbit measured power draw numbers (Table 9) with the previous estimations in Table IV of [32], reveals the previous estimations to be overestimates.

The power use of the OPU does not differ much between buffering and software compression (Table 9). The most power can be conserved by reducing the time spent in phases 3 and 4. The time spent in phase 3 can be drastically reduced by implementing compression for multiple capture modes on the FPGA. The most energy can be conserved by optimizing phase 4. This can be achieved for example by updating the on-board compression algorithms that feature higher compression ratios and thus reduce data volume. For example, using lossy compression methods or dimensionality-reduction-based algorithms. Based on

downlinked highly compressed data, less than 5–6 images per day could be selected to be buffered to the PC and downlinked fully. This could increase the number of images that can be recorded per day, which would increase the average energy use per orbit.

Most other missions outsource implementation of key payload components like Phi-Sat or HICO. However in HYPSO-1's case, the payload software is developed entirely in-house. The authors consider this key for continuous development and in-orbit update of the payload software.

With experience from operating the satellite and scheduling captures, which is the subject of a different publication, the authors envision a more simple, modular and extensible on-board processing pipeline framework using functionality provided by the Linux OS. This more flexible approach would implement every pipeline stage as its own application and have them integrated into a pipeline via an on-ground generated and uplinked shell script. Developing a script generator for the pipeline shell scripts hides the complexity of pure shell scripting, reduces user errors, and can guarantee valid configurations. A potential disadvantage is that this framework might require the designer to put more work into making pipeline modules intercompatible and defining their input and output data format explicitly, since the handover of the data is no longer built into a monolithic pipeline application.

The results of Ethernet throughput testing in Figure 8 are unexpected, and still unexplained. The result of transmission with almost no packet loss around 420–440 Mbit/s in Figure 8 was consistent across tests, but since the packet loss percentage does not reach exactly 0%, picking a frame rate to achieve this data rate could still lead to dropped frames and thus irregular sampling, making processing of the data difficult.

*Conclusions*

A new hardware and software design of a hyperspectral imaging payload with on-board processing capabilites is presented. The design is used in the HYPSO-1 hyperspectral remote sensing satellite. HYPSO-1 continues to reliably record data demonstrating the success of the robust hyperspectral imaging payload OPU design using low-cost COTS components. The software performed reliably with few unexpected issues. Little changes in the architecture are necessary for the future planned satellites.

The presented tests have aided configuring and scheduling satellite operations. The method has characterized the capability of the OPU and aided optimal planning for various operational modes for use in the HYPSO-1 satellite.

HYPSO-2, with identical OPU, but revised software version, is being finalized and is planned for launch in summer 2024. For the planned HYPSO-3 satellite, the SoC will likely be upgraded to a Kria K26 SoM, featuring faster and more CPU cores and a larger FPGA, four-times more DRAM and also full 1 Gbit/s ethernet performance for better frame rate support at higher resolutions.

**Author Contributions:** Conceptualization, D.D.L., M.O. and T.A.J.; methodology, D.D.L., M.O., S.B., R.B. and J.L.G.; software, D.D.L. and S.B.; validation, D.D.L., S.B. and R.B.; investigation, D.D.L. and M.O.; resources, M.O., R.B., T.A.J. and A.J.S.; data curation, D.D.L., S.B. and J.L.G.; writing—original draft preparation, D.D.L., M.O., T.A.J. and A.J.S.; writing—review and editing, D.D.L., M.O., S.B., R.B., J.L.G., T.A.J. and A.J.S.; visualization, D.D.L.; supervision, T.A.J. and A.J.S.; project administration, T.A.J. and A.J.S.; funding acquisition, T.A.J. and A.J.S. All authors have read and agreed to the published version of the manuscript.

**Funding:** This work was supported by the Research Council of Norway through the Centre of Excellence funding scheme NTNU AMOS (grant No. 223254), MASSIVE (grant No. 270959), HYPSCI (grant No. 325961), Green-Platform (grant No. 328724) and ARIEL (grant No. 333229), EEA NO Grants 2014–2021 under Project ELO-Hyp (contract No. 24/2020), the Norwegian Space Agency and the European Space Agency through PRODEX (No. 4000132515).

**Data Availability Statement:** No data is made available.

**Acknowledgments:** The authors acknowledge the work by numerous bachelor and master students to develop, integrate and support the work presented in this article. In particular Joar Gjersund [74], Magne Hov [75], Magnus Danielsen [76], Christoffer Boothby [77], Simen Netteland [78], and Aksel Danielsen [79].

**Conflicts of Interest:** The funders had no role in the design of the study; in the collection, analyses, or interpretation of data; in the writing of the manuscript; or in the decision to publish the results.

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
