# Peer review of "Robust and Reconfigurable On-Board Processing for a Hyperspectral Imaging Small Satellite"

_remotesensing, doi:10.3390/rs15153756_

Round 1

Reviewer 1 Report

In general the document is well written. However some acronyms must be defined. The following points must be addressed before publication:

In the introduction, Line 41 to 46, additional references to discuss the limitations of on-board processing must be included. For example, the following paper:

  1. Salazar, C.; Gonzalez-Llorente, J.; Cardenas, L.; Mendez, J.; Rincon, S.; Rodriguez-Ferreira, J.; Acero, I.F. Cloud Detection Autonomous System Based on Machine Learning and COTS Components On-Board Small Satellites. Remote Sens. 2022, 14, 5597. https://doi.org/10.3390/rs14215597
  2. Park, J.H.; Inamori, T.; Hamaguchi, R.; Otsuki, K.; Kim, J.E.; Yamaoka, K. RGB Image Prioritization Using Convolutional Neural Network on a Microprocessor for Nanosatellites. Remote Sens. 2020, 12, 3941. https://doi.org/10.3390/rs12233941

In Section 2, Line 113-119. Describe how many bands are included in Hyperspectral images?

In Section 3, Line 217. Please indicate where the flight heritage is coming from, use a reference.

Describe the acronyms. For example in Fig.2, what is PPS? 

Include a reference for the CubeSat Space Protocol (CSP), which version was used?

Section 4.

Line 509: 

Please explain how was obtained the equation for the time required for software compression.  What is LSB?

The main point to improve the paper is the discussion about the power of the on-board processing unit.

An important limitation of on-board processing is power and energy consumption. Please include a brief description of power consumption of the designed hardware. Specially, how many watts are required and during how long according to the results presented.

Author Response

Thank you for the review and suggestions! The authors agree on all points.

A sentence with references about the processing power and energy use limitations of small satellite platforms are added.

A sentence saying approximately how many bands hyperspectral data usually contains is added.

References indicating flight heritage are added, as well as a more precise formulation.

A paragraph is added explaining the acronyms used in figure 2 that were previously undefined.

The reference to the CSP source code repository and mention of which version is used is added in the text.

A sentence is added to explain how the equation for computation time was determined.
LSB was referring to the hyperspectral data cube dimensions "lines", "samples" and "bands". Explicit multiplication symbols are added to make the equation more clear, as well as a mention in text. Also, two errors in the labelling of figure 10a are corrected.

The nominal power consumption in Watt of the payload is added to Section 3.2, and a brief note of HYPSO-1's in orbit power use is added to section 4.4.

Reviewer 2 Report

This paper presents the design, implementation, and in-flight demonstration of the onboard processing pipeline of the HYPSO-1 cube-satellite. It is beneficial for learning more details about HYPSO-1. However, the main contribution of this paper can be summarized in Section 1. In addition, Section 2 can be removed.

Author Response

Thank you for the review! The summary of the main contribution is added to the introduction as the second to last paragraph. We disagree however about removing section 2, unless there are specific suggestions you would like us to consider. Additionally, based on another review, specific information about hyperspectral imaging, which Section 2 contains, is recommended.

Reviewer 3 Report

The paper presents a very clear discussion of the hardware and software architecture that led to reliable on-orbit operation of HYPSO-1's On-Board Processing Unit. However, based on the title, I was expecting more information on issues that are specific to hyperspectral imaging. Other than the overall motivation, as explained in the background section, and the use of the CCSDS 123 compression algorithm, the methods described are applicable to reliability in general. I suggest a modification of the title to reflect this, such as "Robust and Reconfigurable Architecture in Support of Hyperspectral Imaging Processing On-Board a Small Satellite."

I look forward to future publications on the subjects of on-board georeferencing, image segmentation, and model training.

Minor Corrections

Line 298: A complete list of service threads is shown in found in Table 2

Line 317: ueye -> uEye (also in Figs 5 and 10)

352: acan -> can

676: word order problem?

Author Response

Thank you for the review and suggestions!

We agree on the point about title, it is changed accordingly.

All mentioned minor corrections are added. 

Round 2

Reviewer 1 Report

We appreciate the effort you have put into revising your manuscript based on the reviewers' comments. However, the revised version lacks clarity in showcasing the implemented modifications. I strongly recommend that the authors mark up any changes made in the manuscript, as indicated.

After carefully examining the changes made by the authors in response to the reviewers' comments, I regret to inform you that the authors have not addressed a crucial reviewer comment regarding an important limitation of on-board processing: power and energy consumption.

Considering the significance of power and energy consumption in on-board processing, I strongly urge the authors to address this limitation and incorporate the requested details in their revised manuscript. Providing this information is vital to comprehensively evaluate the design's feasibility and assess its potential impact.

The manuscript states, "The nominal power consumption of the payload is approximately 2.4 W during processing and 4.3 W during imaging." However, it lacks information on how these measurements were obtained, which raises questions about their accuracy and reliability. 

 Furthermore, the statement "This corresponds to a daily payload energy use of about 9 Wh" does not fully address the important discussion of energy consumption per orbit. Understanding the energy consumption per orbit is essential for determining whether the on-board processing unit (OPU) can sustain continuous operation throughout a single orbit.

Author Response

Thank you for your review!

I am unsure about how to properly submit a new version of a manuscript with the modifications highlighted compared to the previous version, or where it is indicated to do so. The submission website only states to submit a new version of the manuscript.

I can explicitly mention the most important new additions. Section 4.4.1 is added, with a new Figure and a new Table (Figure 11 and Table 9). In section 5, the lines 601 and 621 are added.

I see your point about needing to be more rigorous about the measurement of the energy use of the payload. New paragraphs are added discussing in detail the energy and power aspects of the payload during image recording and on-board processing. The approximate values of power and energy use are now replaced by quantitative measurements obtained from the telemetry of the on-board Electronic Power System (EPS).

In addition, the approximate daily energy use number is converted to a quantitative average energy use per orbit figure that is also derived from the EPS telemetry. The figure can only be average, as HYPSO-1 is not imaging during every orbit by design.